



# Arctic Ocean acidification over the 21st century co-driven by anthropogenic carbon increases and freshening in the CMIP6 model ensemble

Jens Terhaar[1,2], Olivier Torres[3], Timothée Bourgeois[4], and Lester Kwiatkowski[5]

[1]Climate and Environmental Physics, Physics Institute, University of Bern, Bern, Switzerland
[2]Oeschger Center for Climate Change Research, University of Bern, Bern, Switzerland
[3]LMD/IPSL, Ecole Normale Supérieure/PSL Université, CNRS, Ecole Polytechnique, Sorbonne Université, Paris, France
[4]NORCE Norwegian Research Centre and Bjerknes Centre for Climate Research, Bergen, Norway
[5]LOCEAN/IPSL, Sorbonne Université, CNRS, IRD, MNHN, Paris, France

**Correspondence:** Jens Terhaar (jens.terhaar@climate.unibe.ch)

**Abstract.** The uptake of anthropogenic carbon ($C_{ant}$) by the ocean leads to ocean acidification, causing the reduction of pH and the calcium carbonate saturation states of aragonite ($\Omega_{arag}$) and calcite ($\Omega_{calc}$). The Arctic Ocean is particularly vulnerable to ocean acidification due to its naturally low pH and saturation states and due to ongoing freshening and the concurrent reduction in alkalinity in this region. Here, we analyse ocean acidification in the Arctic Ocean over the 21st century across

14 Earth System Models (ESMs) from the latest Coupled Model Intercomparison Project Phase 6 (CMIP6). Compared to the previous model generation (CMIP5), the inter-model uncertainty of projected end-of-century Arctic Ocean $\Omega_{arag/calc}$ is reduced by 44-64%. The strong reduction in projection uncertainties of $\Omega_{arag/calc}$ can be attributed to compensation between $C_{ant}$ uptake and alkalinity reduction in the latest models. Specifically, ESMs with a large increase in Arctic Ocean $C_{ant}$ over the 21st century tend to simulate a relatively weak concurrent freshening and alkalinity reduction, while ESMs with a small increase in $C_{ant}$

simulate a relatively strong freshening and concurrent alkalinity reduction. Although both mechanisms contribute to Arctic Ocean acidification over the 21st century, the increase in $C_{ant}$ remains the dominant driver. Even under the low-emissions shared socioeconomic pathway SSP1-2.6, basin-wide averaged $\Omega_{arag}$ undersaturation occurs before the end of the century. While under the high-emissions pathway SSP5-8.5, the Arctic Ocean mesopelagic is projected to even become undersaturated with respect to calcite. An emergent constraint, identified in CMIP5, which relates present-day maximum sea surface densities

in the Arctic Ocean to the projected end-of-century Arctic Ocean $C_{ant}$ inventory, is found to generally hold in CMIP6. However, a coincident constraint on Arctic declines in $\Omega_{arag/calc}$ is not apparent in the new generation of models. This is due to both the reduction in $\Omega_{arag/calc}$ projection uncertainty and the weaker direct relationship between projected changes in Arctic Ocean $C_{ant}$ and $\Omega_{arag/calc}$. In CMIP6, models generally better simulate maximum sea surface densities in the Arctic Ocean and consequently the transport of $C_{ant}$ into the Arctic Ocean interior, with simulated historical increases in $C_{ant}$ in improved

agreement with observational products.



# 1 Introduction

Human activities such as the burning of fossil fuels, cement production, and land use change have released large amounts of carbon into the atmosphere that cause global warming. The ocean mitigates global warming by taking up around one quarter

of this anthropogenic carbon ($C_{ant}$) (Friedlingstein et al., 2019). However, the increase of carbon in the ocean causes ocean acidification, a process that decreases pH, carbonate ion ($CO_3^{2-}$) concentrations, and in consequence the calcium carbonate ($CaCO_3$) saturation states, of calcite and aragonite minerals (Haugan and Drange, 1996; Orr et al., 2005). The Arctic Ocean is particularly vulnerable to ocean acidification due to its naturally high dissolved inorganic carbon concentrations, low carbonate ion concentrations and its thus naturally low saturation states (Orr et al., 2005; Fabry et al., 2009; Gattuso and Hansson, 2011;

Riebesell et al., 2013; AMAP, 2018).

In contrast to most of the global ocean, Arctic Ocean acidification is not solely caused by increasing $C_{ant}$ concentrations (Anderson et al., 2010; Ulfsbo et al., 2018; Terhaar et al., 2020a, b) but also by freshening (Koenigk et al., 2013; Nummelin et al., 2016; Shu et al., 2018; Brown et al., 2020; Woosley and Millero, 2020). Freshwater from rivers, precipitation, and sea ice typically has much lower $A_T$ and $C_T$ concentrations than the ocean and therefore, in the absence of indirect impacts on other

fluxes, dilutes both marine $A_T$ and $C_T$ (Xue and Cai, 2020). As freshwater $A_T$ and $C_T$ concentrations are generally similar, freshwater fluxes into the ocean act to reduce the difference between $A_T$ and $C_T$, which is approximately equal to the $CO_3^{2-}$ concentration (Waldbusser and Salisbury, 2014; Wanninkhof et al., 2015; Xue and Cai, 2020). Freshening thus reduces marine $CO_3^{2-}$ concentrations and enhances ocean acidification (Bates et al., 2009; Bates and Mathis, 2009; Yamamoto-Kawai et al., 2011). In the Arctic Ocean, projected freshening over the 21$^{st}$ century is larger than in most other ocean regions due to ongoing

sea ice melt, positive precipitation minus evaporation, and large river runoff (Rawlins et al., 2010; Rudels, 2015; Shu et al., 2018).

Due to freshening and increasing $C_{ant}$ concentrations, the Arctic Ocean is projected to be the first large-scale ocean region to become undersaturated with respect to the metastable $CaCO_3$ polymorph aragonite ($\Omega_{arag} < 1$) (Steinacher et al., 2009). Under the RCP8.5 high-emissions scenario, Arctic Ocean mesopelagic waters may even become undersaturated with respect to the

more stable $CaCO_3$ polymorph calcite ($\Omega_{calc} < 1$) before 2100 (Terhaar et al., 2020a). Aragonite and calcite undersaturation is likely to affect the growth, reproduction, and survival of calcifying organisms, such as sea butterflies (Comeau et al., 2010) and foraminifera (Davis et al., 2017), and could have ramifications for the wider Arctic ecosystem (Armstrong et al., 2005; Karnovsky et al., 2008) including some of its most iconic predators, such as gray whales or walruses (Bates et al., 2009).

Projections of Arctic Ocean ocean acidification over the 21$^{st}$ century had considerable subsurface uncertainties in the simu-

lations conducted as part of the Coupled Model Intercomparison Project Phase 5 (CMIP5) (Steiner et al., 2013; Terhaar et al., 2020a), with projected end-of-century basin-wide $\Omega_{arag}$ in mesopelagic waters ranging from 0.61 to 1.05. This large uncertainty has been attributed to multiple factors including variable inflow of Atlantic waters and their subsequent subduction, difficulties resolving the narrow passages between the Arctic Ocean and its surrounding basins and differences in brine rejection during sea ice formation, which is critical to the formation of dense Arctic waters (Terhaar et al., 2019b).





## 1.1 Emergent constraints on Arctic Ocean carbon uptake and acidification


Emergent constraints are a suite of statistical techniques that relate observable trends or sensitivities across multi-model ensembles to differences in model projections in order to reduce future uncertainties (Allen and Ingram, 2002; Hall and Qu, 2006; Hall et al., 2019). Such approaches have been applied extensively within the earth sciences to constrain projections, amongst others, of climate sensitivity (Caldwell et al., 2018), Arctic sea ice (Hall and Qu, 2006), precipitation extremes (O'Gorman, 2012; DeAngelis et al., 2016), carbon cycle feedbacks (Cox et al., 2013; Wenzel et al., 2014), and marine primary production (Kwiatkowski et al., 2017).


Recently Terhaar et al. (2020a) showed that an emergent constraint could be applied to CMIP5 projections of the Arctic Ocean $C_{ant}$ inventory and coincident acidification over the twenty-first century. As the $C_{ant}$ increase in the Arctic Ocean is mainly driven by the inflow of $C_{ant}$-rich waters from the Atlantic and their subsequent subduction in the Barents Sea (Midttun, 1985; Rudels et al., 1994, 2000; Jeansson et al., 2011; Smedsrud et al., 2013), the capability of each model to form dense surface waters in the Barents Sea was shown to strongly influence the future Arctic Ocean $C_{ant}$ inventory. By constraining simulated surface water densities with observations, uncertainties related to the end-of-century Arctic Ocean $C_{ant}$ inventory in 2100 were reduced by around one third and the best estimate under RCP8.5 was increased by 20% to $9.0\pm1.6$ Pg C (Terhaar et al., 2020a). Along with the projected $C_{ant}$ inventory, uncertainties in the projected associated basin-wide Arctic Ocean acidification could also be reduced in CMIP5. It should be noted however, that in CMIP5, projected freshening and reductions in alkalinity were of minor importance for Arctic Ocean acidification over the 21[st] century. Moreover, the models have been shown to underestimate historical freshwater fluxes (1992–2012) in the Arctic Ocean by around 50% (Shu et al., 2018), which suggests they might also have underestimated freshwater fluxes over the 21[st] century.



Given that emergent constraints in many cases conflict with one another (Caldwell et al., 2018; Brient, 2020) and can even be derived from data mined pseudocorrelations (Caldwell et al., 2014), it is critical to test published constraints, and the mechanisms that underpin them, across ESM generations (Eyring et al., 2019; Hall et al., 2019). The CMIP6 simulations provide such an opportunity (Schlund et al., 2020).


## 1.2 From CMIP5 to CMIP6 models and simulations

During the transition from CMIP5 to CMIP6, ESMs have generally improved the simulation of ocean dynamics and marine biogeochemistry (Séférian et al., 2020). Across most ESMs, the horizontal and/or vertical resolution of ocean models has increased, which potentially has large effects on the representation of Arctic Ocean circulation, sea ice dynamics (Docquier et al., 2019), and the carbon cycle (Terhaar et al., 2019b). Ocean biogeochemical model components in CMIP6 also tend to have a more complex representation of the carbon and nutrient cycles than in CMIP5. In particular, the treatment of organic matter carbon cycling has generally evolved, with sedimentation now explicitly simulated in 10 out of 14 ESMs. These developments will likely have a large effect on simulating the Arctic Ocean biogeochemistry given that 50% of the Arctic Ocean is made up of shelf seas (Jakobsson, 2002), where sedimentation and sediment remineralization are crucial components of the carbon and nutrient cycle (Brüchert et al., 2018; Grotheer et al., 2020). Furthermore, the external carbon and nutrient sources from





glaciers, atmospheric deposition, and rivers are represented in more models in CMIP6. Riverine inputs in particular have been shown to be of importance for present-day Arctic Ocean acidification (Anderson et al., 2010; Tank et al., 2012) and its future

changes (Terhaar et al., 2019a).

In this study, we extend recent CMIP6 ocean biogeochemical assessments (e.g. Séférian et al. (2020); Kwiatkowski et al. (2020)) and previous attempts to constrain projected Arctic Ocean $C_{ant}$ uptake by,

1. Assessing projections of the Arctic Ocean $C_{ant}$ inventory over the 21st century in CMIP6 simulations,

2. Exploring the role of $C_{ant}$ inventory increases and freshening in driving concurrent basin-wide ocean acidification in the
Arctic basin,

3. Revaluating previous emergent constraints on the Arctic Ocean $C_{ant}$ inventory and associated acidification using the CMIP6 model ensemble and multiple future emissions scenarios.

## 2  Methods

### 2.1  Arctic Ocean

The Arctic Ocean was defined as the water north of the Fram Strait, the Barents Sea Opening, the Bering Strait and the Baffin Bay following Bates and Mathis (2009). This is consistent with the previously published emergent constraint on projected Arctic Ocean $C_{ant}$ and acidification (Terhaar et al., 2020a).

### 2.2  Earth System Models

An ensemble of 14 ESMs from the Coupled Model Intercomparison Project Phase 6 (CMIP6) (Table 1) was used with one
ensemble member per model. For each model, monthly 3D-fields of dissolved inorganic carbon, total alkalinity, dissolved inorganic phosphorus and silicon, temperature and salinity were used. All 3D-fields were regridded to the regular 1° × 1° grid with 33 depth levels used in the GLobal Ocean Data Analysis Project Version 2 (GLODAPv2) observational product (Lauvset et al., 2016) to add simulated changes of these variables over the 21st century to observations of the present day mean state (see below).

$C_{ant}$ was defined as the difference between dissolved inorganic carbon in the historical (1850–2014) simulations merged with the respective Shared Socioeconomic Pathway (SSP1-2.6, SSP2-4.5, SSP3-7.0, and SSP5-8.5) (2015–2100) (Riahi et al., 2017), and the concurrent pre-industrial control simulations of each model. Output from 13 models was available for SSP1-2.6, output from 12 models for SSP2-4.5 and SSP3-7.0, and output from 14 models for SSP5-8.5 (Table 1).

Changes in total alkalinity ($A_T$) over the 21st century were calculated by subtracting changes in the pre-industrial control
simulations from changes in the respective SSP. To quantify the effect of freshening on changes in $A_T$, the $A_T$ anomalies for each model were further decomposed into changes resulting from freshening and from the combined effect of other bio-





geochemical processes by calculating the temporal evolution of salinity corrected alkalinity with a reference salinity of 35 following Lovenduski et al. (2007).

Ocean carbon chemistry variables in ESMs commonly exhibit mean state biases (Orr et al., 2005; Steiner et al., 2013).

Therefore, observations of $C_T$, $A_T$, dissolved inorganic phosphorus and silicon, temperature and salinity from GLODAPv2 (Lauvset et al., 2016), which is normalized to the year 2002, were used to calculate present day $\Omega_{arag/calc}$, pH and $pCO_2$ using the *mocsy2.0* routine (Orr and Epitalon, 2015) and the equilibrium constants recommended for best practices (Dickson et al., 2007). Future $\Omega_{arag/calc}$, pH and $pCO_2$ were calculated for each model as the sum of the simulated changes in $C_T$, $A_T$, dissolved inorganic phosphorus and silicon, temperature and salinity from 2002 onwards and the observed quantities in 2002.

The present-day maximum sea surface density in the Arctic Ocean was calculated from monthly climatologies over 1986-2005, constructed from the respective salinity and temperature outputs of each model. Maximum sea surface density was calculated, as in Terhaar et al. (2020a), as the mean density of the densest 5% of Arctic surface waters (95[th] percentile waters) over all 12 months of the year.

## 2.3 Simulations

The simulations performed within CMIP5 were not forced with the same atmospheric $CO_2$ concentrations as the simulations performed under CMIP6. In CMIP5, historically observed atmospheric $CO_2$ concentrations were used from 1850 to 2005 (Meinshausen et al., 2011). From 2006 onwards, the $CO_2$ concentrations follow the different RCPs. In CMIP6, the historical period was extended until 2014 and thereafter $CO_2$ concentrations follow the different SSPs (Meinshausen et al., 2019).

The different land and energy use assumptions in the SSPs (Riahi et al., 2017) compared to the RCPs lead to higher atmo-
spheric $CO_2$ trajectories over the 21[st] century for the Tier 1 SSPs compared to their RCP counterparts (O'Neill et al., 2016), which results in globally greater surface ocean acidification in CMIP6 compared to CMIP5 (Kwiatkowski et al., 2020). Furthermore, historical atmospheric $CO_2$ concentrations were also refined. This refinement did not change the average atmospheric $CO_2$ concentration from 1850 to 2005 ($\Delta pCO_2^{atm} = 0\pm1$ ppm), but changed annual $CO_2$ concentration for single years by up to $\pm2$ ppm.

## 140 2.4 $C_{ant}$ scaling

The different atmospheric $CO_2$ trajectories over the 21[st] century between CMIP5 and CMIP6 complicates a comparison of simulated Arctic Ocean $C_{ant}$ inventories between model generations. To nevertheless compare the simulated Arctic Ocean $C_{ant}$, we used the commonly applied scaling approach that assumes that the change in marine $C_{ant}$ is proportional to the atmospheric $C_{ant}$ concentration (Gloor et al., 2003; Mikaloff Fletcher et al., 2006; Gerber et al., 2009; Gruber et al., 2009). Under this
assumption, the $C_{ant}$ inventory in 2100 for each scenario (RCP8.5, SSP1-2.6, SSP2-4.5, SSP3-7.0) is linearly rescaled to that of SSP5-8.5 by multiplying the simulated Arctic Ocean $C_{ant}$ inventory under the respective scenario by the ratio of the mean atmospheric $C_{ant}$ concentration from 1850 to 2100 in SSP5-8.5 and the respective scenario.





This approximation is likely imprecise when very different scenarios are compared, such as SSP1-2.6 and SSP5-8.5, as the effects of circulation changes, sea ice melt and warming are not considered. However, when comparing scenarios with the same radiative forcing, such as SSP5-8.5 and RCP8.5, it permits a first order comparison.

### 2.5 Observational constraints

As for the ESMs, the maximum sea surface density was calculated based on a monthly sea surface density climatology on a regular $1° \times 1°$ grid, which was constructed from observed monyhly salinity and temperature climatologies in the World Ocean Atlas 2018 (Locarnini et al., 2018; Zweng et al., 2018).

The density uncertainty was calculated from the temperature and salinity uncertainties that were reported by the World Ocean Atlas following standard propagation of uncertainty. The total uncertainty is a combination of (1) the standard deviations for sea surface density derived from published standard deviations of sea surface temperature and salinity for each grid cell and each month in the World Ocean Atlas 2018, and (2) the standard deviation from the weighted mean of the 95[th] percentile density waters (Terhaar et al., 2020a).

### 2.6 Probability density functions of $C_{ant}$

The probability density functions (PDFs) for unconstrained projections of the $C_{ant}$ inventory were calculated using equal weights for each model and assuming a Gaussian distribution.

The PDFs for the constrained projections of the $C_{ant}$ inventory were calculated as the convolution of the observational PDF and the PDF that results from the emergent relationship between present-day maximum sea surface density and projected $C_{ant}$ following previous studies (Cox et al., 2013; Wenzel et al., 2014; Kwiatkowski et al., 2017).

Extending the analysis of Terhaar et al. (2020a), PDFs for the constrained projections of the $C_{ant}$ inventory were calculated not only for the year 2100 but for each year from 2002 to 2100 and not only for the highest emission scenario (RCP8.5) but for the four SSPs (SSP1-2.6, SSP2-4.5, SSP3-7.0, and SSP5-8.5).

## 3 Results

### 3.1 Arctic Ocean $C_{ant}$ inventory

#### 3.1.1 Multi-model mean

Over the historical period from 1850 to 2005, the CMIP6 ESMs simulate an Arctic Ocean $C_{ant}$ increase of $2.1 \pm 0.6$ Pg C (inter-model standard deviation; Fig. 1a). As such, the CMIP6 ESMs simulate an Arctic Ocean $C_{ant}$ inventory in 2005 that is 61% higher than the inventory that was simulated by the previous model generation in CMIP5 ($1.3 \pm 0.7$ Pg C; Terhaar et al. (2020a)). This is still 36% below the data-based estimate for the period from 1765 to 2005 of $3.3 \pm 0.3$ Pg C (Terhaar et al., 2020b).



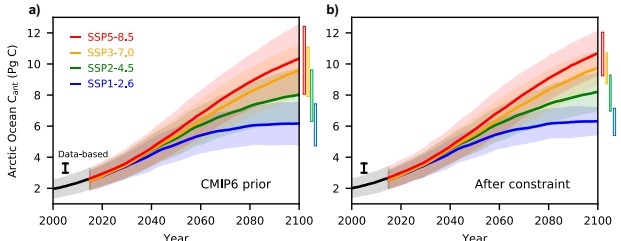

**Figure 1.** Arctic Ocean $C_{ant}$ inventory over the 21st century. **a)** Time-series of multi-model mean Arctic Ocean $C_{ant}$ inventory over the 21st century for SSP1-2.6 (blue), SSP2-4.5 (green), SSP3-7.0 (orange), and SSP5-8.5 (red) with the respective standard deviation across the model ensemble (shading) (n=12-14). The bars on the right side of panels indicate the inter-model standard deviation in year 2100. **b)** Time-series of Arctic Ocean $C_{ant}$ inventory for each scenario after the emergent constraint is applied.

Over the 21st century the Arctic Ocean $C_{ant}$ inventory increases depending on the SSP. Following the low-emission pathway SSP1-2.6 leads to a projected Arctic Ocean $C_{ant}$ inventory in 2100 of $6.2 \pm 1.3$ Pg C. With increasing atmospheric $C_{ant}$ concentrations from SSP1-2.6 to SSP5-8.5, the projected Arctic Ocean $C_{ant}$ inventory in 2100 also increases, resulting in 8.0

$\pm$ 1.7 Pg C for SSP2-4.5, $9.6 \pm 1.6$ Pg C for SSP3-7.0, and $10.3 \pm 2.2$ Pg C for SSP5-8.5. The relatively large uncertainties across the model ensemble (17-21%) result in an overlap of the simulated $C_{ant}$ inventories in 2100 for SSP2-4.5, SSP3-7.0, and SSP5-8.5 within $\pm$ one standard deviation.

### 3.1.2 Constrained results

As was shown for RCP8.5 in the CMIP5 model ensemble (Terhaar et al., 2020a), a linear relationship between maximum sea

surface density and the Arctic Ocean $C_{ant}$ inventory in 2100 is found across the CMIP6 model ensemble for all four of the SSPs ($r^2$=0.63-0.72; Fig. 2a,c,e,g). By deriving a similar relationship for the projected $C_{ant}$ inventory in all years from 2000 to 2100 and combining this with observations of present-day sea surface density, the uncertainty of the projected Arctic Ocean $C_{ant}$ inventory can be reduced throughout the 21st century (Fig. 1b). The emergent relationship is significant over all years from 2015 to 2100 ($p<0.05$) and the $r^2$ increases from 0.39 in 2014 to 0.63-0.72 in 2100 depending on the scenario. In the year

2100, this results in Arctic Ocean $C_{ant}$ inventory estimates of $6.3 \pm 0.9$ Pg C (SSP1-2.6), $8.2 \pm 1.2$ Pg C (SSP2-4.5), $9.8 \pm 1.1$ Pg C (SSP3-7.0), and $10.7 \pm 1.4$ Pg C (SSP5-8.5) (Fig. 2b,d,f,h). As such, the emergent constraint is shown to slightly increase the CMIP6 multi-model mean projected $C_{ant}$ inventory for each SSP (+2 to +4 %) and substantially reduce associated uncertainties (-29 to -31 %) resulting in greater separation of the SSPs (Fig. 1).





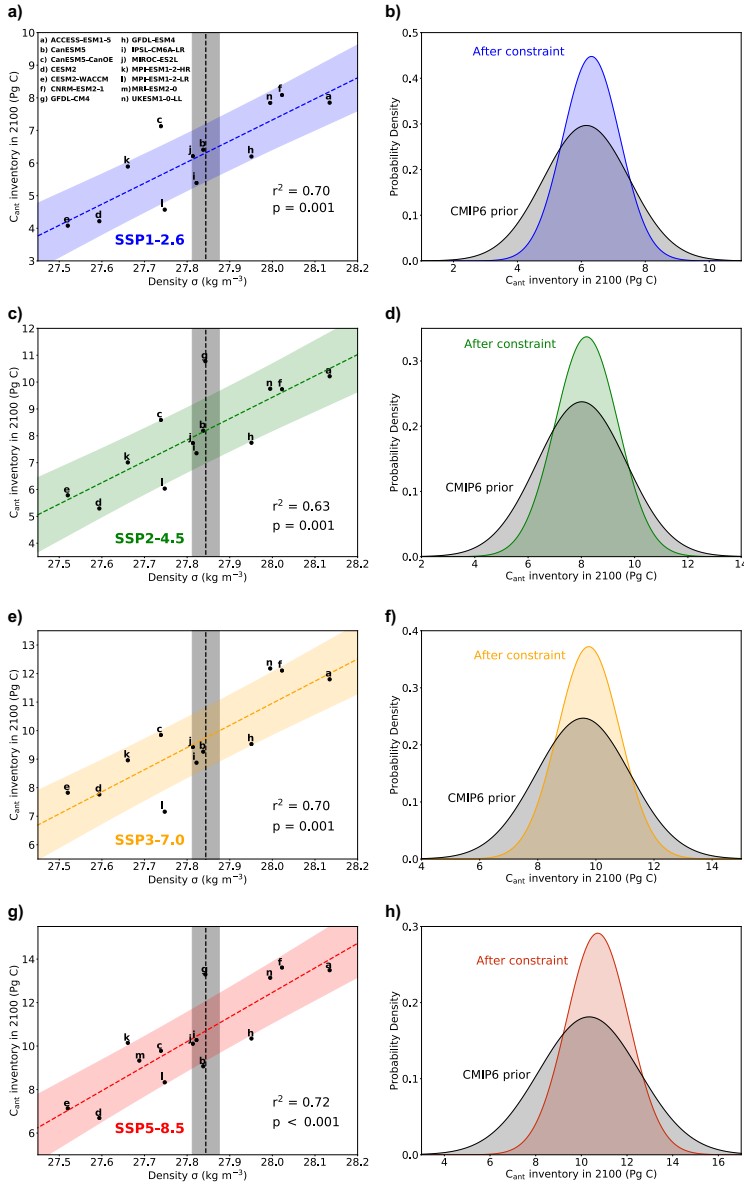

**Figure 2.** Emergent constraint on Arctic Ocean $C_{ant}$ inventory in 2100. Maximum Arctic Ocean sea surface density (95[th] percentile) and Arctic Ocean $C_{ant}$ inventory in 2100 for 12-14 ESMs for **a)** SSP1-2.6 (blue), **c)** SSP2-4.5 (green), **e)** SSP3-7.0 (orange), and **g)** SSP5-8.5 (red). The ordinary least squares regressions (dashed lines) and the $\pm$ one $\sigma$ prediction intervals (shaded area) are shown for each SSP. Probability density functions before and after applying the emergent constraint for **b)** SSP1-2.6, **d)** SSP2-4.5, **f)** SSP3-7.0, and **h)** SSP5-8.5.

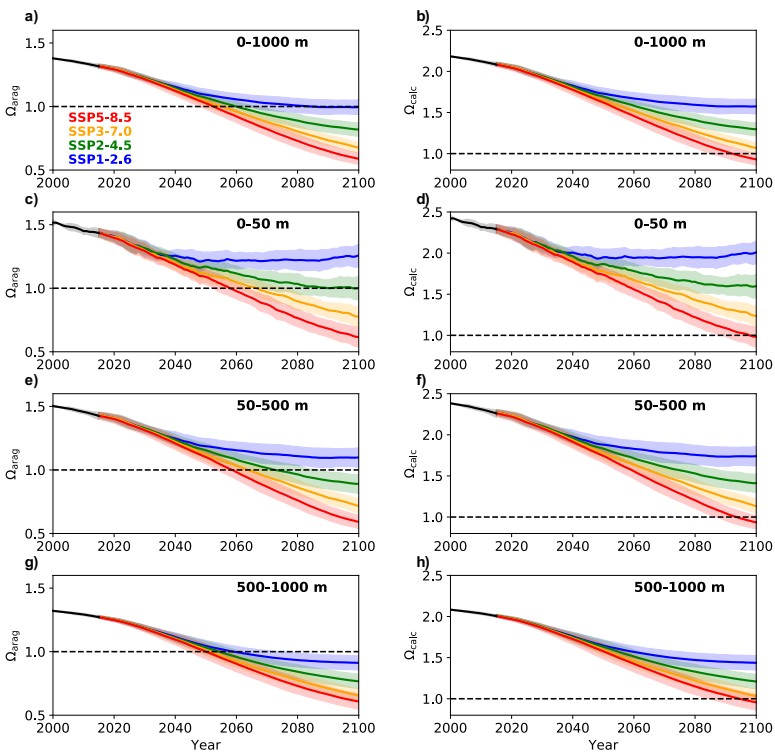

**Figure 3.** Basin-wide Arctic Ocean calcium carbonate saturation states over the 21st century. Time series of multi-model mean basin-wide saturation states of aragonite from 2000 to 2014 (black) and from 2015 to 2100 for SSP1-2.6 (blue), SSP2-4.5 (green), SSP3-7.0 (orange), and SSP5-8.5 (red) averaged **a)** from 0 to 1000 m, **c)** from 0 to 50 m, **e)** from 50 to 500 m, and **g)** from 500 to 1000 m with $\pm$ one standard deviation (n=12-14) shown as shaded area. The differences when changing from historical simulations to the SSPs result from the different number of ensemble members that are available for each simulation. **b), d), f), h)** the same time series but for calcite.

## 3.2 Ocean acidification

### 3.2.1 Multi-model mean

Over the 21st century, ocean acidification leads to a reduction in $\Omega_{arag}$ and $\Omega_{calc}$ (Fig. 3). Until around 2040, the reduction of both saturation states averaged over the first 1000 m is independent of the SSP. From 2040 onwards, the rate of the reduction depends on the pathway and by 2100 basin-wide $\Omega_{arag}$ averaged over the first 1000 m reaches $1.00 \pm 0.06$ (SSP1-2.6), $0.82 \pm 0.05$ (SSP2-4.5), $0.66 \pm 0.04$ (SSP3-7.0), and $0.59 \pm 0.05$ (SSP5-8.5), while $\Omega_{calc}$ reaches $1.56 \pm 0.09$ (SSP1-2.6), $1.29 \pm 0.08$ (SSP2-4.5), $1.06 \pm 0.06$ (SSP3-7.0), and $0.92 \pm 0.05$ (SSP5-8.5).



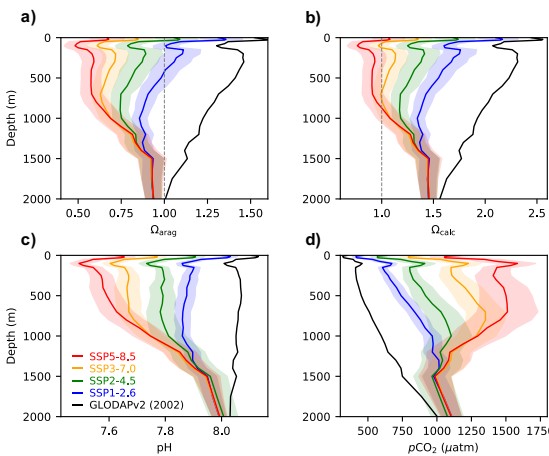

**Figure 4.** Vertical profile of basin-wide Arctic Ocean calcium carbonate saturation states, pH and $p\text{CO}_2$ in 2100. Multi-model mean vertical profiles of **a)** aragonite and **b)** calcite saturation states, **c)** pH, and **d)** $p\text{CO}_2$ in 2100 for SSP1-2.6 (blue), SSP2-4.5 (green), SSP3-7.0 (orange), and SSP5-8.5 (red) averaged **a)** with $\pm$ one standard deviation (n=12-14) shown as shaded area. Observed profiles in 2002 from GLODAPv2 are shown in black (Lauvset et al., 2016)

The first 1000 m of the Arctic Ocean are projected to be on average undersaturated with respect to aragonite by 2100 under all of the SSPs with $\Omega_\text{arag}$ in SSP1-2.6 just below 1 (0.995). Vertically, the entire water column will be undersaturated with respect to aragonite in 2100 for all pathways but SSP1-2.6 (Fig 4). Under this low-emissions pathway, Arctic Ocean waters above 500 m are projected to remain supersaturated with respect to aragonite, while those below 500 are projected to be undersaturated. For the more stable calcium carbonate polymorph calcite, basin-wide Arctic Ocean undersaturation is only projected under SSP5-8.5. Under this high-emissions pathway, basin-wide undersaturation is projected for the water masses between 50 and 800 m.

Alongside declines in calcium carbonate saturation states, pH and $p\text{CO}_2$ are also projected to change over the 21$^\text{st}$ century. Basin-wide averaged pH in the first 1000 m decrease from 8.06 in 2002 to 7.88±0.02 (SSP1-2.6), 7.79 $\pm$ 0.03 (SSP2-4.5), 7.68 $\pm$ 0.03 (SSP3-7.0), and 7.61 $\pm$ 0.03 (SSP5-8.5), while basin-wide averaged $p\text{CO}_2$ in the first 1000 m increases to 739±38 $\mu$atm (SSP1-2.6), 918 $\pm$ 75 $\mu$atm (SSP2-4.5), 1209 $\pm$ 92 $\mu$atm (SSP3-7.0), and 1428 $\pm$ 124 $\mu$atm (SSP5-8.5). The lowest pH values and highest $p\text{CO}_2$ values are projected to occur between 100 and 1000 m (Fig. 4).

### 3.2.2 Emergent constraints on acidification

The emergent relationship between observed maximum sea surface density and end-of-century Arctic Ocean acidification that was previously identified in CMIP5 models (Terhaar et al., 2020a) does not exist anymore in the new CMIP6 model generation (Fig. 5). Despite the emergent constraint approach still functioning for projections of the Arctic Ocean $\text{C}_\text{ant}$ inventory in CMIP6, this no longer translates into an emergent constraint for Arctic Ocean $\Omega_\text{arag/calc}$. However, if all changes in ocean



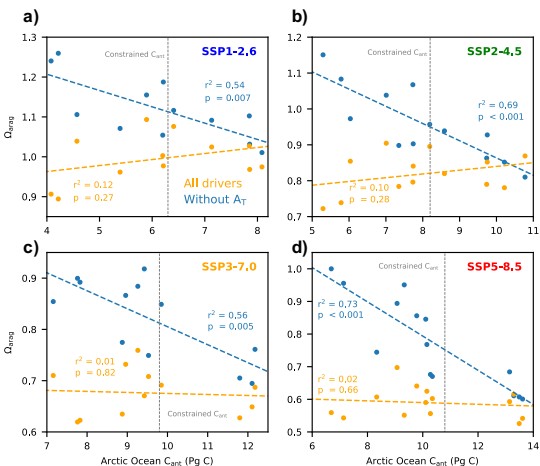

**Figure 5.** The CMIP6 ensemble relationship between end-of-century $\Omega_{arag}$ and the $C_{ant}$ inventory in the Arctic Ocean. Basin-wide $\Omega_{arag}$ averaged over the first 1000 m with (orange) and without (blue) taking into account concurrent changes in alkalinity ($A_T$) against the Arctic Ocean $C_{ant}$ inventory in year 2100 for **a)** SSP1-2.6, **a)** SSP2-4.5, **c)** SSP3-7.0, and **d)** SSP5-8.5.

biogeochemistry with the exception of alkalinity, which is kept constant at observed levels from 2002 (Lauvset et al., 2016), are considered when estimating changes in $\Omega_{arag/calc}$, emergent relationships are still found between maximum sea surface

density and $\Omega_{arag}$ across all SSPs ($r^2 = 0.54$-$0.73$). When decreases in alkalinity ($A_T$) are also taken into account, $\Omega_{arag/calc}$ decrease further and the emergent relationship disappears ($r^2 = 0.01$-$0.12$).

### 3.3 Comparison between CMIP5 and CMIP6

Compared to ESMs from CMIP5, the new generation of ESMs (CMIP6) has improved in simulating the maximum Arctic Ocean sea surface density. Specifically, negative density biases have been reduced and the inter-model range in maximum sea

surface density has substantially decreased from 3.6 kg m$^{-3}$ in CMIP5 (Terhaar et al., 2020a) to 0.9 kg m$^{-3}$ in CMIP6 (Fig. 2). As a result, the inter-model range of the $C_{ant}$ inventory in the CMIP6 model ensemble is also reduced (Fig. 2). Moreover, without the negative maximum sea surface density bias, the simulated multi-model mean Arctic Ocean $C_{ant}$ inventory in 2005 is 61% higher than the inventory that was simulated by the previous model generation ($1.3 \pm 0.7$ Pg C) (Terhaar et al., 2020a).

At the end of the 21$^{st}$ century, the unconstrained simulated Arctic Ocean $C_{ant}$ inventory under SSP5-8.5 is 37% larger and the

230 uncertainty is 19% smaller than the unconstrained simulated $C_{ant}$ inventory in 2100 under RCP8.5 ($7.5 \pm 2.9$ Pg C) (Terhaar et al., 2020a). After applying the constraint, the Arctic Ocean $C_{ant}$ inventory in 2100 under SSP5-8.5 ($10.7 \pm 1.4$ Pg C) is 19% larger than the constrained Arctic Ocean $C_{ant}$ inventory under RCP8.5 ($9.0 \pm 1.6$ Pg C). This difference is of the same order of magnitude as the difference in prescribed atmospheric $CO_2$ concentration over the 21$^{st}$ century, which is higher in SSP5-8.5



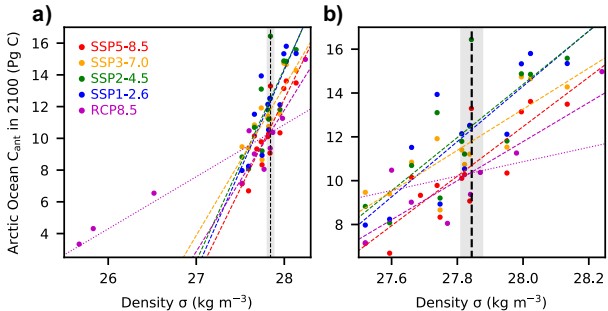

**Figure 6.** Arctic Ocean $C_{ant}$ inventory in 2100 against maximum Arctic Ocean sea surface density (95[th] percentile) for SSP1-2.6 (blue; $r^2$=0.70), SSP2-4.5 (green; $r^2$=0.63), SSP3-7.0 (orange; $r^2$=0.70), SSP5-8.5 (red; $r^2$=0.72), and RCP8.5 (magenta; $r^2$=0.74) **a)** over the entire density range and **b)** over the density range from 27.5 to 28.3 kg m$^{-3}$. The simulated Arctic Ocean $C_{ant}$ inventory in 2100 for each scenario was rescaled to SSP5-8.5 using the mean atmospheric $C_{ant}$ concentration from 1850 to 2100 as a scaling factor. Linear fits use all available models for each scenario (dashed lines). For RCP8.5 an additional fit is shown excluding the three models with density anomalies below 27.5 kg m$^{-3}$ **b)** (dotted line; $r^2$=0.79).

(CMIP6) than RCP8.5 (CMIP5) (Meinshausen et al., 2011, 2019) and therefore results in greater surface ocean acidification

for approximately the same radiative forcing (Kwiatkowski et al., 2020).

To compare the emergent constraint across scenarios with different atmospheric $CO_2$ concentrations, the simulated Arctic Ocean $C_{ant}$ inventory in 2100 for each scenario was rescaled to SSP5-8.5 using the mean atmospheric $C_{ant}$ concentration from 1850 to 2100 as a linear scaling factor (Fig. 6). The relationship remains robust ($r^2$=0.63–0.74) for all five analysed scenarios. The slope of the emergent relationship is however substantially steeper in CMIP6 (9.4–12.6 Pg C kg$^{-1}$ m$^3$) than in CMIP5

(3.3 Pg C kg$^{-1}$ m$^3$). However, the slope in CMIP5 increases to 8.9 Pg C kg$^{-1}$ m$^3$ if the three CMIP5 models with particularly low maximum sea surface densities (<27.5 kg m$^{-3}$) are excluded (dotted line in Fig. 6). The resulting constrained estimate for the rescaled Arctic Ocean $C_{ant}$ inventory decreases from the low-emission scenario to the high-emission scenario from 12.3 to 10.7 Pg C. When comparing the two high-emission scenarios, the rescaled Arctic Ocean $C_{ant}$ inventories are 10.7 Pg C (SSP5-8.5) and 10.4 Pg C (RCP8.5). The latter remains unchanged if the three CMIP5 models with particularly low maximum

sea surface densities (<27.5 kg m$^{-3}$) are excluded.

## 4 Discussion

### 4.1 Arctic Ocean $C_{ant}$ inventory

Across the CMIP6 model ensemble, the Arctic Ocean $C_{ant}$ storage over the 21[st] century is highly related to maximum sea surface densities (Fig. 2), which predominately occur in the Barents Sea (Midttun, 1985; Smedsrud et al., 2013; Terhaar et al.,

2020a). The inter-model range in maximum sea surface density in the Barents Sea is mainly explained by differences in sea





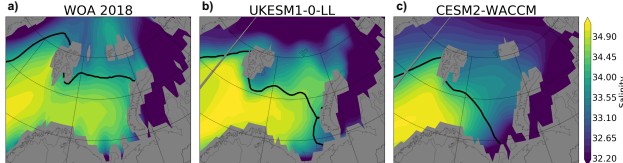

**Figure 7.** Annual mean sea surface salinity in the Barents Sea and its neighbouring waters from **a)** the World Ocean Atlas 2018, **b)** the CMIP6 model with the highest (UKESM1-0-LL) and **c)** lowest (CESM2-WACCM) maximum sea surface density and salinity. The $0°$C isotherm indicates approximately the extent of Atlantic waters (black solid line).

surface salinities ($r^2$=0.93), which are influenced by brine rejection (Midttun, 1985; Årthun et al., 2011) and the strength of inflowing, saltier Atlantic waters through the Barents Sea Opening (Fig. 7). Compared to CMIP5 models, the reduced negative bias of simulated maximum sea surface densities (Fig. 6) indicates model improvement in simulating the circulation of Atlantic and Arctic surface waters. Despite the reduced inter-model range, the robust relationship between maximum sea surface density and $C_{ant}$ across model generations (Fig. 6) supports evidence that inflowing Atlantic waters through the Barents Sea Opening and their transformation into deep and intermediate waters via brine rejection are the dominant process governing Arctic Ocean $C_{ant}$ increases (Midttun, 1985; Rudels et al., 1994, 2000; Jeansson et al., 2011; Smedsrud et al., 2013). The application of observational constraints to this emergent multi-model relationship in order to constrain the projected Arctic Ocean $C_{ant}$ inventory (Fig. 2) and focus model development therefore remains promising.

However, the slope of the linear relationship between maximum sea surface density and the Arctic Ocean $C_{ant}$ inventory over the 21$^{st}$ century in the CMIP5 model ensemble ($3.3 \pm 0.6$ Pg C kg$^{-1}$ m$^3$, scaled to SSP5-8.5 atmospheric $C_{ant}$ concentrations) is three to four times less than that in the CMIP6 model ensemble ($12.6 \pm 2.6$ Pg C kg$^{-1}$ m$^3$ for SSP1-2.6, $12.1 \pm 2.8$ Pg C kg$^{-1}$ m$^3$ for SSP2-4.5, $9.4 \pm 1.9$ Pg C kg$^{-1}$ m$^3$ for SSP3-7.0, and $11.3 \pm 2.0$ Pg C kg$^{-1}$ m$^3$ for SSP5-8.5) (Fig. 6). The reduced slope in the CMIP5 ensemble is mainly caused by three models with maximum surface density anomalies well below 27.5 kg m$^{-3}$. Excluding these three models, the remaining CMIP5 models follow a slope of $8.9 \pm 2.2$ Pg C kg$^{-1}$ m$^3$, in broad agreement with the CMIP6 model ensemble (Fig. 6). This suggests that the linear emergent relationship does not hold below a certain value of maximum sea surface density below which the impact on deep-water formation and subsequent $C_{ant}$ storage in the Arctic Ocean is limited. However, as the two linear relationships happen to cross the observed maximum sea surface density in nearly the same location (Fig. 6), the constrained $C_{ant}$ inventory for the CMIP5 model ensemble remains almost entirely unchanged when the three low-density models are excluded.

Even without the low-density bias in the Barents Sea, the constrained Arctic Ocean $C_{ant}$ inventory in 2005 in CMIP6 remains 36% below the data-based estimate (Terhaar et al., 2020b). This underestimation is partly due to the different definition of $C_{ant}$ in data-based estimates and ESMs. While the historical simulations in CMIP5 and CMIP6 typically start in 1850, data-based estimates account for all $C_{ant}$ since 1765. This leads to an underestimation of the global ocean $C_{ant}$ inventory by ESMs of around 30% (Bronselaer et al., 2017) and of around 20% in the Arctic Ocean (Terhaar et al., 2019b). Even if we increased the constrained Arctic Ocean $C_{ant}$ inventory in 2005 by 20%, an underestimation of around 16% remains compared to the data-





based estimate. This underestimation of the data-based estimate suggests that all ESMs are missing additional pathways of $C_{ant}$ entry into the Arctic Ocean, other than the principal pathway via the Barents Sea. Indeed, small-scale density flows along continental slopes can be observed in different regions of the Arctic Ocean (Rudels et al., 1994; Jones et al., 1995) but cannot

be simulated by the coarse resolution of most ESMs. As such, the constrained estimates of the Arctic Ocean $C_{ant}$ inventory presented here are likely still a lower boundary.

Recent observation of dilution of $A_T$ and $C_T$ in surface waters of the Amerasian basins caused by freshening have led to the hypothesis that continuous freshening might turn the Arctic Ocean from a sink of $C_{ant}$ into a source over the 21[st] century (Woosley and Millero, 2020). However, observations in the Eurasian basins, which receive more saline Atlantic water and

less freshwater input, still show increases of $C_{ant}$ concentrations (Ulfsbo et al., 2018) over a depth of 1500 m in the last 20 years. The CMIP5 and CMIP6 model ensembles both simulate continuous accumulation of $C_{ant}$ in the Arctic Ocean under all SSPs (Fig. 1), suggesting that the subduction of $C_{ant}$-rich Atlantic waters in the Barents Sea remains larger than any loss of $C_{ant}$ in surface waters over the 21[st] century. Nevertheless, the reduction of the storage rate of $C_{ant}$ under SSP5-8.5 (Fig. 1) in combination with constantly increasing atmospheric $CO_2$ concentrations (Riahi et al., 2017) indicates that dilution may reduce

the capacity of the Arctic Ocean to store further $C_{ant}$ as suggested by Woosley and Millero (2020).

## 4.2   Arctic Ocean acidification

### 4.2.1   $\Omega_{arag/calc}$, pH, and $p\mathrm{CO_2}$ in 2100

Even under the most optimistic scenario assessed (SSP1-2.6), the Arctic Ocean will become on average undersaturated with respect to aragonite with possible consequences for calcifying organisms (Comeau et al., 2010) and the food chain (Armstrong

et al., 2005; Karnovsky et al., 2008). In this scenario only water masses above 500 m remain supersaturated this century. These findings are in good agreement with idealized estimates of $\Omega_{arag/calc}$ that project the future $C_{ant}$ inventory based on the Transient Time Distribution method and observed CFC-12 concentrations (Anderson et al., 2010; Terhaar et al., 2020b).

In addition to widespread Arctic Ocean undersaturation with respect to aragonite, water masses between 50 and 800 m are even projected to become undersaturated with respect to calcite by the end of the century under SSP5-8.5. Calcite undersat-

uration in Arctic Ocean subsurface waters will likely further enhance the pressure on the Arctic Ocean ecosystem as calcite forming organisms, such as foraminifera (Davis et al., 2017) and coccolithophores (Kottmeier et al., 2016), experience potential impacts on growth and survival.

Furthermore, the projected increases in $p\mathrm{CO_2}$ (Fig. 4c, d) alongside projected increases in its seasonal amplitude (McNeil and Sasse, 2016; Kwiatkowski and Orr, 2018) is likely to lead to hypercapnic conditions that might directly affect the growth

and survival of Arctic fish (Frommel et al., 2012; Schmidt et al., 2017; Kunz et al., 2018) under high emissions scenarios.



### 4.2.2 Driving processes of acidification

The CMIP6 model ensemble shows astonishingly good agreement with respect to projections of Arctic Ocean acidification over the 21$^{st}$ century. Compared to the CMIP5 model ensemble, the uncertainties in projected $\Omega_{arag}$ averaged over the first 1000 m in 2100 have been reduced from 0.13 to 0.04–0.06 and those for $\Omega_{calc}$ from 0.21 to 0.05–0.09.

The main reason for the reduced uncertainty appears to be that the ESMs that have a lower maximum sea surface density and that thus take up less $C_{ant}$ in the Arctic Ocean over the 21$^{st}$ century (Fig. 2) are the same models that simulate a stronger reduction in $A_T$ (Fig. 8). To explain this negative correlation between $C_{ant}$ uptake and reductions in $A_T$, we propose the following mechanism: Over the 21$^{st}$ century, melting of sea ice, land ice, and increased river runoff are projected to freshen the Arctic Ocean (Koenigk et al., 2013; Nummelin et al., 2016; Shu et al., 2018) and to reduce $A_T$ (Fig. 8) (Woosley and Millero, 315 2020). In the CMIP6 ensemble, this freshening and reduction of $A_T$ tends to be stronger in models with lower sea surface salinities and densities, i.e. the models that simulate less inflow of saline Atlantic waters such as CESM2-WACCM (Fig. 7), have lower Barents Sea surface salinity and density, weak deep-water formation, and therefore less $C_{ant}$ storage (Fig. 2). In contrast, models with a larger inflow of saline Atlantic water have stronger deep-water formation, greater $C_{ant}$ storage, and a smaller freshening and reduction in $A_T$. This compensation then results in a similar simulated reduction of $\Omega_{arag/calc}$ over the 320 21$^{st}$ century across all models in the CMIP6 ensemble, either through a reduction in $A_T$ or an increase in $C_T$. Other possible drivers, such as changes in temperature and salinity, are likely of minor importance given the relatively good correlation between the projected $C_{ant}$ inventory and basin-wide reduction of $\Omega_{arag/calc}$ over the 21$^{st}$ century when $A_T$ is not taken into account ($r^2$=0.54–0.73).

    In the CMIP5 model ensemble, this compensation effect did not exist and projected saturation states and pH were mainly 325 driven by the projected $C_{ant}$ storage in each model (Terhaar et al., 2020a) with changes in $A_T$, temperature, and salinity of minor importance. The relatively small changes in $A_T$ in CMIP5 compared to CMIP6 might have been caused by an underestimation of Arctic Ocean freshening over the 21$^{st}$ century given that the CMIP5 models also underestimated historical Arctic Ocean freshening by around 50% in the Arctic Ocean (Shu et al., 2018). In the CMIP6 model ensemble this negative bias with respect to freshwater fluxes and subsequent dilution of $A_T$ appears to have been reduced. This is probably a consequence of better 330 representation of riverine fluxes as well as increased model resolution and hence improved circulation (Séférian et al., 2020).

    To estimate the likely impacts of $C_T$ increases and $A_T$ decreases to Arctic Ocean acidification over the 21$^{st}$ century, one can assess their relative contributions to acidification at constrained estimates of end-of-century $C_{ant}$ (dotted line in Fig 5). In SSP5-8.5 for example, increasing $C_T$ along with changes in temperature and salinity reduce $\Omega_{arag}$ from 1.38 to 0.75, while decreasing $A_T$ further reduces this to 0.59. Thus, the future storage of $C_{ant}$ in the Arctic Ocean still remains the main driver of 335 Arctic Ocean acidification, with acidification further enhanced by decreases in $A_T$.

## 5 Conclusions

Earth System Models tend to have improved their performance in the Arctic Ocean from CMIP5 to CMIP6. The negative bias with respect to present-day maximum sea surface density is substantially reduced in CMIP6 and the historical $C_{ant}$ inventory





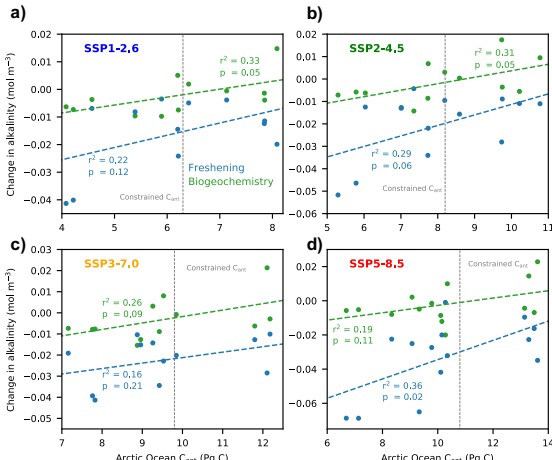

**Figure 8.** CMIP6 basin-wide changes in alkalinity ($A_T$) averaged over the first 1000 m from 2002 to 2100 caused by fresh water dilution (blue) and changes in biogeochemistry (green) against the Arctic Ocean $C_{ant}$ inventory for **a)** SSP1-2.6, **b)** SSP2-4.5, **c)** SSP3-7.0, and **d)** SSP5-8.5. $A_T$ anomalies were decomposed by calculating the temporal evolution of salinity corrected alkalinity following Lovenduski et al. (2007).

is closer to the data-based estimate (Terhaar et al., 2020b). Furthermore, the inter-model range of projected $\Omega_{arag/calc}$ and the

$C_{ant}$ inventory have been reduced. Uncertainties in the future $C_{ant}$ inventory under four shared socioeconomic pathways can be further reduced by applying the previously identified emergent constraint between present-day maximum Arctic Ocean sea surface density and the future $C_{ant}$ inventory (Terhaar et al., 2020a).

     Hall et al. (2019) define "confirmed" emergent constraints as fulfilling the following criteria: 1) a plausible mechanism, 2) verification of that mechanism, and 3) out-of-sample testing. The mechanism underpinning the relationship between maximum

sea surface densities and the projected $C_{ant}$ inventory in the Arctic Ocean is consistent with observations and has been verified in an ocean-only model at different resolutions (Terhaar et al., 2020a). The testing of this emergent constraint in the CMIP6 ensemble can be considered out-of-sampling, although the extent of model independence between generations remains questionable (Knutti et al., 2013; Sanderson et al., 2015). Despite the overall emergent constraint on projected Arctic Ocean $C_{ant}$ being similar between CMIP6 and CMIP5, the change in the multi-model emergent relationship indicates that further valida-

tion is required. Specifically, it appears that the linear relationship between maximum sea surface densities and the projected $C_{ant}$ inventory breaks down in models with extremely low-biased sea surface densities, highlighting a potential limitation to this emergent constraint that requires further assessment.

     While the mechanisms underlying emergent constraints on the future Arctic Ocean $C_{ant}$ inventory appear to be consistent between CMIP5 and CMIP6, this is not the case for the previously identified emergent constraint on ocean acidification vari-

ables ($\Omega_{arag/calc}$, pH, and $p$CO$_2$). In the CMIP6 model ensemble, projected model uncertainties in $\Omega_{arag/calc}$, pH, and $p$CO$_2$ are



dramatically reduced in CMIP5 and are not only driven by the increase in $C_{ant}$, but also by a reduction in $A_T$ due to pan-Arctic freshening. A weak inflow of saline Atlantic waters in ESMs results in lighter surface waters and less $C_{ant}$ storage over the 21$^{st}$ century but also in a stronger reduction of alkalinity caused by a stronger freshening. A strong Atlantic inflow however, appears to limit freshening and reductions in alkalinity but results in greater $C_{ant}$ storage. Although both $C_{ant}$ storage and alkalinity

reductions, contribute considerably to Arctic Ocean acidification, our results suggest that $C_{ant}$ remains the dominant process. In the CMIP5 model ensemble, the influence of freshening on emergent constraints on acidification variables might have been of limited importance because a) freshening was largely underestimated (Shu et al., 2018) and b) differences in simulating the inflow of Atlantic waters through the Barents Sea Opening were considerably larger across the model ensemble (Terhaar et al., 2020a). As such, the absence of an emergent constraint on projected Arctic acidification in CMIP6 could be viewed as a

consequence of successful model development. Indeed, there is clearly no need for such constraints when models are in broad agreement.

Independent of the driving mechanism, the projected Arctic Ocean acidification over the 21$^{st}$ century has possibly grave consequences for the wider Arctic Ocean ecosystem (Gattuso and Hansson, 2011; Riebesell et al., 2013; AMAP, 2018). Under all socio-economic pathways, the upper 1000 m of the Arctic Ocean will be on average undersaturated with respect to aragonite.

As such, keystone species like the sea butterfly (aragonitic pteropod *Limacina helicina*) may lose most to all of their suitable habitat in the Arctic Ocean (Comeau et al., 2010). However, under the low-emissions pathway SSP1-2.6 aragonite undersaturation is projected to be avoided in the upper 500 m of the water column, providing a potential refuge. Under the high-emissions pathway SSP5-8.5, water masses between 50 and 800 m are additionally projected to become undersaturated with respect to calcite. While isolated regions of the surface Arctic Ocean, strongly influenced by riverine inputs, are already seasonally un-

dersaturated with respect to calcite (Bates and Mathis, 2009), subsurface Arctic waters would be the first to exhibit annually averaged and basin-wide calcite undersaturation over a depth of several hundred meters. Vertically migrating organisms that form calcite shells and skeletons, such as coccolithophores (Kottmeier et al., 2016) and foraminifera (Davis et al., 2017), may lose their natural refugia area to which they migrate during the day (Berge et al., 2015). These hostile conditions for calcifying organisms will likely cause their decline with unknown consequences for the wider Arctic food web (Armstrong et al., 2005;

Karnovsky et al., 2008) and its iconic species (Bates et al., 2009).

*Data availability.* The Earth system model output used in this study is available via the Earth System Grid Federation (https://esgf-node. ipsl.upmc.fr/projects/esgf-ipsl/). Observations from the World Ocean Atlas 2013 (https://www.nodc.noaa.gov/OC5/woa18/) and GLODAPv2 (https://www.nodc.noaa.gov/ocads/oceans/GLODAPv2_2019/) are available via the National Oceanic and Atmospheric Administration.

*Author contributions.* The study was led by JT, who made the figures and wrote the initial manuscript. TB and LK provided help for the

analyses and interpretation of the results. OT processed the CMIP6 model data (download, regridding). All authors contributed to the final manuscript.





*Competing interests.* No competing interests are present.

*Acknowledgements.* This project has received funding from the European Union's Horizon 2020 research and innovation programme under grant agreement No 821003 (4C), No 641816 (CRESCENDO), and No 820989 (COMFORT). The work reflects only the authors' view; the
European Commission and their executive agency are not responsible for any use that may be made of the information the work contains. This project has also received funding from the Swiss National Science Foundation (grant no. PP00P2_170687), the Agence Nationale de la Recherche grant ANR-18-ERC2-0001-01 (CONVINCE) and the Research Council of Norway under grant number 275268 (COLUMBIA). We acknowledge the World Climate Research Programme's Working Group on Coupled Modelling, which is responsible for CMIP. For CMIP the US Department of Energy's Program for Climate Model Diagnosis and Intercomparison provided coordinating support and led
the development of software infrastructure in partnership with the Global Organisation for Earth System Science Portals. We also thank the IPSL modelling group for the software infrastructure, which facilitated CMIP6 analysis.



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

**Table 1.** The CMIP6 ESMs used in this study with their ocean-sea ice and marine-biogeochemical (MBG) model components.

| Model | Ocean-sea ice | MBG | Data DOI |
| --- | --- | --- | --- |
| ACCESS-ESM1.5 (Ziehn et al., 2020) | MOM5, CICE4 | WOMBAT | Ziehn et al. (2019a, b) |
| CanESM5 (Swart et al., 2019e) | NEMO 3.4.1-LIM2 | CMOC | Swart et al. (2019a, b) |
| CanESM5-CanOE (Swart et al., 2019e; Christian et al., 2020) | NEMO 3.4.1-LIM2 | CanOE | Swart et al. (2019c, d) |
| CESM2 (Danabasoglu et al., 2020) | POP2-CICE5 | MARBL-BEC | Danabasoglu (2019a, b) |
| CESM2-WACCM (Danabasoglu et al., 2020) | POP2-CICE5 | MARBL-BEC | Danabasoglu (2019c, d) |
| CNRM-ESM2-1 (Séférian et al., 2019) | NEMOv3.6-GELATOv6 | PISCESv2-gas | Seferian (2018a, b) |
| GFDL-CM4[a] (Held et al., 2019; Dunne et al., 2020a) | MOM6, SIS2 | BLINGv2 | Guo et al. (2018a, b) |
| GFDL-ESM4 (Dunne et al., 2020b; Stock et al., 2020) | MOM6, SIS2 | COBALTv2 | Krasting et al. (2018); John et al. (2018) |
| IPSL-CM6A-LR (Boucher et al., 2020) | NEMOv3.6-LIM3 | PISCESv2 | Boucher et al. (2018a, b) |
| MIROC-ES2L (Hajima et al., 2020) | COCO | OECO2 | Hajima et al. (2019); Tachiiri et al. (2019) |
| MPI-ESM1.2-HR (Müller et al., 2018; Mauritsen et al., 2019) | MPIOM | HAMOCC6 | Schupfner et al. (2019); Jungclaus et al. (2019) |
| MPI-ESM1.2-LR (Mauritsen et al., 2019) | MPIOM | HAMOCC6 | Wieners et al. (2019b, a) |
| MRI-ESM2[b] (Yukimoto et al., 2019a) | MRICOM4 | NPZD | Yukimoto et al. (2019b, c) |
| UKESM1-0-LL (Sellar et al., 2019) | NEMO v3.6, CICE | MEDUSA-2 | Tang et al. (2019); Good et al. (2019) |

[a] Only SSP2-4.5 and SSP5-8.5

[b] Only SSP5-8.5