# Peer review of "Arctic Ocean acidification over the 21st century co-driven by anthropogenic carbon increases and freshening in the CMIP6 model ensemble"

_Biogeosciences, 2020_

## Referee Comment (RC1) · Scott C. Doney (Referee) · 11 Jan 2021

Review of: Arctic Ocean acidification over the 21st century co-driven by anthropogenic carbon increases and freshening in the CMIP6 model ensemble Jens Terhaar et al. 2020 Biogeosciences Discussions https://doi.org/10.5194/bg-2020-456

The manuscript presents an analysis of an ensemble of coupled global climate-carbon cycle model simulations for the recent historical period through the end of this century. The models' Arctic Ocean inorganic carbon systems are analyzed to quantify the growth of the anthropogenic carbon inventory and the underlying driving factors related to ocean thermocline ventilation and freshwater/alkalinity trends. The approach follows

on prior studies using an earlier generation of Earth System Models (CMIP5) identifying substantial changes in CMIP6 model dynamics and the performance of a pair of so-called emergent constraints relating changes in carbon uptake and acidification to surface density (a metric of ventilation rates for mesopelagic ocean). Overall this is a well-constructed manuscript both in terms of the underlying analysis methodology and presentation, and this work is relevant to a number of science communities involved in climate and ocean carbon cycle science as well as marine ecology related to the impacts of ocean acidification. My recommendation is for minor edits to address my comments below that are primarily requests for clarifications on the methodology and results in the text.

Line 6 "the inter-model uncertainty of projected end-of-century Arctic Ocean Omega arag/calc"

Should this be the temporal change of saturation state comparing preindustrial minus end of century? This relates to the model bias correction described in Lines 120-125 using the GLODAP data. Would be useful to describe in abstract the data-based bias correction that is applied to the models.

Also need to clarify in Abstract when the text is describing the water-column (or whole Arctic) change or when discussion trends for particular depth levels (e.g., surface, mesopelagic).

Line 18-20 "In CMIP6, models generally better simulate maximum sea surface densities in the Arctic Ocean and consequently the transport of Cant into the Arctic Ocean interior, with simulated historical increases in Cant in improved agreement with observational products."

Perhaps description of improvement in model ocean physics would be better placed in abstract on Line 5 before discussion of carbon system.

Line 42-42 "Due to freshening and increasing Cant concentrations, the Arctic Ocean

is projected to be the first large-scale ocean region to become undersaturated with respect to the metastable $CaCO_3$ polymorph aragonite (Omega arag < 1) (Steinacher et al., 2009)."

Clarify if referring to Steinacher et al. results on trends in Arctic surface ocean or full depth ocean.

Line 55 In the section on emergent constraints, is there any evidence that variations in extent of sea-ice loss affects local air-sea anthropogenic $CO_2$ uptake and thus inventory?

Line 84 "sedimentation now explicitly simulated in 10 out of 14 ESMs"

Does "sedimentation" here refer to gravitational particle sinking? Or does this refer to fluxes at the water-sediment surface? Please clarify.

Line 87 "Furthermore, the external carbon and nutrient sources"

How many of the CMIP6 models include dissolved inorganic carbon and alkalinity concentrations in river fluxes? It would be useful to expand Table 1 or add a Table 2 to display the differences in model treatment of freshwater inorganic carbon chemistry.

Line 104, Section 2.2 Some more detail is needed on the specific models and simulations used for the ocean biogeochemistry CMIP6 ensemble. For example, did all the coupled models follow the protocols outlined in:

Orr et al. 2017: Biogeochemical protocols and diagnostics for the CMIP6 Ocean Model Intercomparison Project (OMIP), Geoscientific Model Development, 10, 2169-2199, doi:10.5194/gmd-10-2169-2017

Also, given the importance of river input into the Arctic, some additional discussion is needed on the model treatment of river freshwater, inorganic and organic carbon, and alkalinity.

Line 115-118 "To quantify the effect of freshening on changes in AT, the AT anomalies

for each model were further decomposed into changes resulting from freshening and from the combined effect of other bio-geochemical processes by calculating the temporal evolution of salinity corrected alkalinity with a reference salinity of 35 following Lovenduski et al. (2007)."

A salinity correction implicitly assumes a freshwater end-member for alkalinity. For the Arctic was a non-zero end-member used to account for non-zero river alkalinity?

Line 241-243 "The resulting constrained estimate for the rescaled Arctic Ocean Cant inventory decreases from the low-emission scenario to the high-emission scenario from 12.3 to 10.7 Pg C."

Perhaps would be useful to clarify again that the rescaled constraint only informs the actual emergent constraint that is in Figure 2.

Line 251 "surface salinitities"

Should be "salinities"

Line 356 "dramatically reduced in CMIP5"

Should this read "dramatically reduced compared to the uncertainties in CMIP5"?
* * *

---

## Referee Comment (RC2) · Claudine Hauri (Referee) · 21 Jan 2021

Terhaar et al., use 14 Earth System Models from CMIP6 to assess Arctic Ocean acidification and how these CMIP6 results compare to earlier results from the CMIP5 generation models. The CMIP6 models project a reduced uncertainty in aragonite and calcite saturation states compared to CMIP5, which is mainly due to compensation of over or underestimation of Canth through over or underestimation of freshening and reduction of alkalinity. Overall, CMIP6 models improved on simulating max. sea surface densities and therefore better simulate the transport of Cant into the Arctic Ocean.

The manuscript is well written. There were a few sections that need some clarifications

(see comments below), but in general I think this manuscript can be accepted with minor revisions.

Page 1: Lines 35 – 37: I think this sentence should be rephrased. This could be misinterpreted to "freshening reduces the difference in a way that carbonate ion concentration will be zero"

Page 2, Line 48. There are two Bates et al., 2009 references in this manuscript, so it is not clear which one the authors refer to with Bates et al., 2009. Also, I doubt Bates is the right reference here since he is a biogeochemist and not a mammal biologist.

Page 2, Line 49: delete "ocean"

Page 3, Lines 55- 61. This first paragraph could be used to explain emergent constraints a bit better. It becomes clear on lines 66 -69, but readers who are not familiar with this methodology could be easily lost in this section.

Lines 69 and 70: This is confusing. I thought emergent constraints were already applied to CMIP5 models in a previous publication (Terhaar et al., 2020a)?

Page 4, Lines 110 – 112: Was the difference based on annual means, monthly means? Would be good to clarify here.

Page 5: Line 137: How were historical atmospheric CO2 concentrations refined? Reference?

Overall, I think the methodologies would become clearer if the authors included a more detailed explanation on how emergent constraints were used on the CMIP6 models. I realize that there was a section on emergent constraints in the introduction, but 1) I found this section hard to follow, and 2) it wasn't clear whether this was only applied to CMIP5 or also CMIP6.

Page 7, Lines 177 -183: Since results from the historical simulation were compared to CMIP5 it might make sense to compare the 21st century results to CMIP 5 as well? Or

even better all together move this comparison to the section dedicated to CMIP5 and CMIP6 comparison later on (see comment below)

Page 10, Line 201: "the first 1000m of the Arctic Ocean" does not make sense. This could be interpreted horizontally or vertically. It also needs to be clarified whether it is from the surface or bottom.

Line 209: reword "first 1000 m"

Line 212: maybe change to "between 100 and 1000 m from the surface"

Page 11, section 3.3 Since you are comparing CMIP5 and CMIP 6 results here, I would get rid of the comparison you do at the beginning of the results to avoid repetition and simplify everything.

Page 17, line 360: first comma does not seem necessary

Line 378: "area" does not seem necessary in combination with refugia.

Line 380: As mentioned earlier, I'm not sure Bates is the right reference here.
* * *

---

## Author Comment (AC1) · 8 Feb 2021

**Response to the reviewers**

We thank the reviewer Prof. Scott Doney for his very helpful and constructive comments. In the following we address the comments point by point.
* * *
The manuscript presents an analysis of an ensemble of coupled global climate-carbon cycle model simulations for the recent historical period through the end of this century. The models' Arctic Ocean inorganic carbon systems are analyzed to quantify the growth of the anthropogenic carbon inventory and the underlying driving factors related to ocean thermocline ventilation and freshwater/alkalinity trends. The approach follows on prior studies using an earlier generation of Earth System Models (CMIP5) identifying substantial changes in CMIP6 model dynamics and the performance of a pair of so-called emergent constraints relating changes in carbon uptake and acidification to surface density (a metric of ventilation rates for mesopelagic ocean). Overall this is a well-constructed manuscript both in terms of the underlying analysis methodology and presentation, and this work is relevant to a number of science communities involved in climate and ocean carbon cycle science as well as marine ecology related to the impacts of ocean acidification. My recommendation is for minor edits to address my comments below that are primarily requests for clarifications on the methodology and results in the text.

**1.1** — Line 6 "the inter-model uncertainty of projected end-of-century Arctic Ocean Omega arag/calc"

Should this be the temporal change of saturation state comparing preindustrial minus end of century? This relates to the model bias correction described in Lines 120-125 using the GLODAP data. Would be useful to describe in abstract the data-based bias correction that is applied to the models.

Also need to clarify in Abstract when the text is describing the water-column (or whole Arctic) change or when discussion trends for particular depth levels (e.g., surface, mesopelagic).

**Reply**: Line 6 will be changed in the revised manuscript to as follows:

" Compared to the previous model generation (CMIP5), the inter-model uncertainty of projected changes over the 21$^{st}$ century in Arctic Ocean $\Omega_{arag/calc}$ averaged over the first 1000 m is reduced by 44-64%."

Lines 12 and 13 will also be changed for clarification to:

"Even under the low-emissions shared socioeconomic pathway SSP1-2.6, basin-wide averaged $\Omega_{arag}$ undersaturation in the first 1000 m occurs before the end of the century."

**1.2** — Line 18-20 "In CMIP6, models generally better simulate maximum sea surface densities in the Arctic Ocean and consequently the transport of Cant into the Arctic Ocean interior, with simulated historical increases in Cant in improved agreement with observational products."

Perhaps description of improvement in model ocean physics would be better placed in abstract on Line 5 before discussion of carbon system.

**Reply**: This will be changed as suggested.

**1.3** — Line 42-42 "Due to freshening and increasing Cant concentrations, the Arctic Ocean is projected to be the first large-scale ocean region to become undersaturated with respect to the metastable CaCO3 polymorph aragonite (Omega arag < 1) (Steinacher et al., 2009)."

Clarify if referring to Steinacher et al. results on trends in Arctic surface ocean or full depth ocean.

**Reply**: Steinacher et al. results refer to the entire watercolumn. This will be clarified in the revised manuscript:

"Due to freshening and increasing $C_{ant}$ concentrations, the Arctic Ocean is projected to be the first large-scale ocean region to become undersaturated with respect to the metastable $CaCO_3$ polymorph aragonite ($\Omega_{arag} < 1$) throughout the entire water column (Steinacher et al., 2009)."

**1.4** — Line 55 In the section on emergent constraints, is there any evidence that variations in extent of sea-ice loss affects local air-sea anthropogenic CO2 uptake and thus inventory?

**Reply**: At present, the air-sea $CO_2$ uptake plays a minor role for the Arctic Ocean $C_{ant}$. Indeed, most of the $C_{ant}$ enters the Arctic Ocean with Atlantic waters, which are already saturated with respect to $C_{ant}$ (Terhaar et al., 2020b). These waters then sink into the deeper ocean in the Barents Sea (Midttun, 65 1985; Rudels et al., 1994, 2000; Jeansson et al., 2011; Smedsrud et al., 2013, Terhaar et al., 2019b).). The strong relationship between the sea surface density in the Barents Sea and the end-of-century Arctic Ocean $C_{ant}$ inventory (Figure 2) indicates this will not change in the near future.

**1.5** — Line 84 "sedimentation now explicitly simulated in 10 out of 14 ESMs" Does "sedimentation" here refer to gravitational particle sinking? Or does this refer to fluxes at the water-sediment surface? Please clarify.

**Reply**: In the revised manuscript the sentence will be changed to:

"In particular, the treatment of organic matter carbon cycling has generally evolved, with remineralization of particles in sediments now simulated in 10 out of 14 ESMs."

**1.6** — Line 87 "Furthermore, the external carbon and nutrient sources" How many of the CMIP6 models include dissolved inorganic carbon and alkalinity concentrations in river fluxes? It would be useful to expand Table 1 or add a Table 2 to display the differences in model treatment of freshwater inorganic carbon chemistry.

**Reply**: The information will be added to Table 1 as suggested.

**1.7** — Line 104, Section 2.2 Some more detail is needed on the specific models and simulations used for the ocean biogeochemistry CMIP6 ensemble. For example, did all the coupled models follow the protocols outlined in:
Orr et al. 2017: Biogeochemical protocols and diagnostics for the CMIP6 Ocean Model Intercomparison Project (OMIP), Geoscientific Model Development, 10, 2169-2199, doi:10.5194/gmd-10-2169-2017

**Reply**: More specific information about the models and simulations,e.g. spin-up length and riverine input, will be added to the revised manuscript as suggested.

**1.8** — Also, given the importance of river input into the Arctic, some additional discussion is needed on the model treatment of river freshwater, inorganic and organic carbon, and alkalinity.

**Reply**: As suggested, the riverine carbon and alkalinity fluxes will be added to Table 1. Furthermore, additional text will be added about the model treatment of riverine freshwater, carbon and alkalinity.

**1.9** — Line 115-118 "To quantify the effect of freshening on changes in AT, the AT anomalies for each model were further decomposed into changes resulting from freshening and from the combined effect of other bio-geochemical processes by calculating the temporal evolution of salinity corrected alkalinity with a reference salinity of 35 following Lovenduski et al. (2007)." A salinity correction implicitly assumes a freshwater end-member for alkalinity. For the Arctic was a non-zero end-member used to account for non-zero river alkalinity?

**Reply**: A zero end-member was assumed for freshwater. This is correct for models with no alkalinity in freshwater but an overestimation for models with alkalinity in freshwater. Unfortunately, the information about alkalinity in freshwater is not available for most models. Moreover, with the available model output it is impossible to quantify the contribution of land ice melt, sea ice melt, precipitation minus evaporation, and riverine input to freshwater changes. For simplicity, we assumed a zero alkalinity end-member. This will be explained and discussed in the revised manuscript.

**1.10** — Line 241-243 "The resulting constrained estimate for the rescaled Arctic Ocean Cant inventory decreases from the low-emission scenario to the high-emission scenario from 12.3 to 10.7 Pg C." Perhaps would be useful to clarify again that the rescaled constraint only informs the actual emergent constraint that is in Figure 2.

**Reply**: As suggested, this will be clarified in the figure and the figure legend.

**1.11** — Line 251 "surface salinitities" Should be "salinities"

**Reply**: This will be changed in the revised manuscript.

**1.12** — Line 356 "dramatically reduced in CMIP5" Should this read "dramatically reduced compared to the uncertainties in CMIP5"?

**Reply**: Yes it should. This will be changed in the revised manuscript.

---

## Author Comment (AC2) · 8 Feb 2021

**Response to the reviewers**

We thank the reviewer Prof. Claudine Hauri for her very helpful and constructive comments. In the following we address the comments point by point.
* * *
Terhaar et al., use 14 Earth System Models from CMIP6 to assess Arctic Ocean acidification and how these CMIP6 results compare to earlier results from the CMIP5 generation models. The CMIP6 models project a reduced uncertainty in aragonite and calcite saturation states compared to CMIP5, which is mainly due to compensation of over or underestimation of Canth through over or underestimation of freshening and reduction of alkalinity. Overall, CMIP6 models improved on simulating max. sea surface densities and therefore better simulate the transport of Cant into the Arctic Ocean. The manuscript is well written. There were a few sections that need some clarifications (see comments below), but in general I think this manuscript can be accepted with minor revisions.

**1.1** — Page 1: Lines 35 – 37: I think this sentence should be rephrased. This could be misinterpreted to "freshening reduces the difference in a way that carbonate ion concentration will be zero"

**Reply**: This has now been changed to:

" As freshwater AT and CT concentrations are generally similar, freshwater fluxes into the ocean typically act to reduce the difference between AT and CT, decreasing marine CO32- concentrations and ocean pH (Waldbusser and Salisbury, 2014; Wanninkhof et al., 2015; Xue and Cai, 2020; Bates et al., 2009; Bates and Mathis, 2009; Yamamoto-Kawai et al., 2011). In the Arctic Ocean, projected freshening..."

**1.2** — Page 2, Line 48. There are two Bates et al., 2009 references in this manuscript, so it is not clear which one the authors refer to with Bates et al., 2009. Also, I doubt Bates is the right reference here since he is a biogeochemist and not a mammal biologist.

**Reply**: The Biogeoscience LaTex template formats the two citations of Bates in 2009 in two different ways: "Bates et al., 2009" and "Bates and Mathis, 2009". If we can distinguish these two citations in a better way, we will do so.

We will remove the reference to Bates et al., 2009 here as suggested by the referee and replace it with Jay et al., 2011 and AMAP, 2018.

**1.3** — Page 2, Line 49: delete "ocean"

**Reply**: "ocean" will be deleted as suggested.

**1.4** — Page 3, Lines 55- 61. This first paragraph could be used to explain emergent constraints a bit better. It becomes clear on lines 66 -69, but readers who are not familiar with this methodology could be easily lost in this section.

**Reply**: The introduction and description of emergent constraints will be expanded in the revised manuscript as suggested.

**1.5** — Lines 69 and 70: This is confusing. I thought emergent constraints were already applied to CMIP5 models in a previous publication (Terhaar et al., 2020a)?

**Reply**: Emergent constraints were indeed applied in a previous publication. That publication (Terhaar et al., 2020a) constrained $C_{ant}$ and acidification. Both applications are important for the Introduction and therefore mentioned in the Introduction. We have decided not to add the reference again in order to improve readability.

**1.6** — Page 4, Lines 110 – 112: Was the difference based on annual means, monthly means? Would be good to clarify here.

**Reply**: The difference is based on annual means. The sentence will be changes in the revised manuscript to as follows:

"$C_{ant}$ was defined as the difference between annual means of dissolved inorganic carbon in the historical (1850–2014) simulations merged with the respective Shared Socioeconomic Pathway (SSP1-2.6, SSP2-4.5, SSP3-7.0, and SSP5-8.5) (2015–2100) (Riahi et al., 2017), and the concurrent pre-industrial control simulations of each model."

**1.7** — Page 5: Line 137: How were historical atmospheric CO2 concentrations refined? Reference?

**Reply**: In CMIP6, the historical global annual mean $CO_2$ concentrations were updated with additional data available since CMIP5. Furthermore, global $CO_2$ concentrations were additionally provided as monthly latitudinally resolved concentrations with model groups free to choose the forcing files they use. This will be detailed in the revised manuscript with the appropriate reference provided.

**1.8** — Overall, I think the methodologies would become clearer if the authors included a more detailed explanation on how emergent constraints were used on the CMIP6 models. I realize that there was a section on emergent constraints in the introduction, but 1) I found this section hard to follow, and 2) it wasn't clear whether this was only applied to CMIP5 or also CMIP6.

**Reply**: We will try to make this introductory section on emergent constraints easier to follow. In addition, the following details on how we applied emergent constraint techniques will be added to the Methods in the revised manuscript:

"To calculate the emergent constraint, first an ordinary least squares regression was calculated between the simulated present-day maximum sea surface density and the Arctic Ocean $C_{ant}$ inventory in 2100 for each ESM of the CMIP6 model ensemble. The uncertainty range was estimated using the 1-$\sigma$ prediction interval. In a second step the probability density functions (PDFs) from the observations was convoluted with the PDF from the linear regression, assuming a Gaussian distribution in both cases. The convolution of both PDFs is the constrained projection of the Arctic Ocean $C_{ant}$ inventory following previous studies (Cox et al., 2013; Wenzel et al., 2014; Kwiatkowski et al., 2017)."

**1.9** — Page 7, Lines 177 -183: Since results from the historical simulation were compared to CMIP5 it might make sense to compare the 21st century results to CMIP 5 as well? Or even better all together move this comparison to the section dedicated to CMIP5 and CMIP6 comparison later on (see comment below)

**Reply**: This will be moved as suggested by the referee.

**1.10** — Page 10, Line 201: "the upper 1000m of the Arctic Ocean" does not make sense. This could be interpreted horizontally or vertically. It also needs to be clarified whether it is from the surface or bottom.

**Reply**: This will be changed in the revised manuscript from "the first" to "the upper 1000 m".

**1.11** — Line 209: reword "first 1000 m"

**Reply**: This will be changed in the revised manuscript to "the upper 1000 m".

**1.12** — Line 212: maybe change to "between 100 and 1000 m from the surface"

**Reply**: This will be changed as suggested.

**1.13** — Page 11, section 3.3 Since you are comparing CMIP5 and CMIP 6 results here, I would get rid of the comparison you do at the beginning of the results to avoid repetition and simplify everything.

**Reply**: This will be changed as suggested by the referee.

**1.14** — Page 17, line 360: first comma does not seem necessary

**Reply**: The comma will be deleted.

**1.15** — Line 378: "area" does not seem necessary in combination with refugia.

**Reply**: "area" will be deleted in the revised manuscript.

**1.16** — Line 380: As mentioned earlier, I'm not sure Bates is the right reference here.

**Reply**: We will remove the reference to Bates et al., 2009 here as suggested by the referee and replace it with Jay et al., 2011 and AMAP, 2018.